# Appropriate complementary feeding practice and associated factors among mothers of children aged 6–23 months in Bhimphedi rural municipality of Nepal

Thag Bahadur Gurung[1], Rajan Paudel[2], Anil K. C.[3]*, Aashish Acharya[4], Pravin Kumar Khanal[5]

1 Environment Health and Health Care Waste Management Section, Management Division, Department of Health Service, Kathmandu, Nepal, 2 Central Department of Public Health, Institute of Medicine, Tribhuvan University, Kathmandu, Nepal, 3 Smart Health Global, Kathmandu, Nepal, 4 Manmohan Memorial Institute of Health Sciences, Kathmandu, Nepal, 5 Health Office, Lamjung, Nepal

* kcanil707@gmail.com

## Abstract

### Background

Appropriate complementary feeding plays a crucial role in the enhancement of child survival; and promotes healthy growth and development. Evidence has shown that appropriate complementary feeding is effective in preventing malnutrition and child mortality. Thus, the main objective of this study is to assess the prevalence of appropriate complementary feeding practice and associated factors among mothers of children aged 6–23 months.

### Methodology

A community-based cross-sectional study was conducted from August to December 2018. A total of 259 mothers who had children aged 6–23 months were selected randomly from the 714 eligible mothers. A structured questionnaire was used to collect the data from the respondents. The data were collected in a tablet phone-based questionnaire using the Open Data Kit mobile application by face-to-face interview. Data analysis was done in SPSS version 21. Multivariable logistic regression was used to identify the factor associated with appropriate complementary feeding practice.

### Result

The prevalence of appropriate complementary feeding practice was 25%. Mother and father with formal education (AOR 6.1, CI: 1.7–22.4 and AOR 5.6 CI: 1.5–21.2 respectively), counseling on IYCF (AOR 4.2, CI: 1.5–12.3), having kitchen garden (AOR 2.4, CI: 1.1–5.2) and food secured family (AOR 3.0, CI: 1.0–8.9) had higher odds of appropriate complementary feeding practice.

**Data Availability Statement:** All relevant data are within the manuscript and its supporting

information files or can be found in https://doi.org/10.6084/m9.figshare.24633153.v1.

**Funding:** The author(s) received no specific funding for this work.

**Competing interests:** The authors have declared that no competing interests exist.

## Conclusion

This study revealed that a significant proportion of mothers had inappropriate complementary feeding practice for their children aged 6–23 months. This study highlights the need for behavior change communication and promotion of kitchen garden to address the associated factors and promote appropriate complementary feeding practice.

## Introduction

After the age of six months, breast milk is not sufficient for the growth and development of the baby. The starting of other liquids and food along with breast milk to meet the nutritional requirement for the growth and development of the baby is known as complementary feeding [1, 2]. Appropriate child feeding has a crucial role in the enhancement of child survival; and promotes healthy growth and development, a productive future generation, and national development. Ensuring optimal nutrition during the first 2 years of a child's life is fundamental to lowering morbidity and mortality, minimizing the risks of chronic diseases, and promoting better development in general [3]. Exclusive breastfeeding up to 6 months of age and breastfeeding up to 12 months of age occupied the number one position with the introduction of appropriate complementary feeding starting at 6 months of age holding number three in terms of their effectiveness in preventing child mortality. Almost, one-fifth of under-five mortality due to diarrhea and ARI were estimated to be prevented by these two interventions alone [4]. Undernutrition resulting from poor complementary feeding practice is responsible for 35% of under-five mortality [5].

South Asia region has the lowest rate of minimum meal frequency (47%), minimum dietary diversity (23%), and minimum acceptable diet (13%) in comparison to other regions [6]. Nepal Demographic and Health Survey Report 2016 revealed that only 46% of children aged 6–23 months received meals with the minimum recommended diversity), 71% received meals at the minimum frequency, and 36% met the criteria of a minimum acceptable diet [7].

A handful of studies were conducted in Nepal regarding the indicators of Infant and Young Child Feeding (IYCF) and associated factors. One study revealed that more than half of the children timely initiated complementary feeding at 6 months (57.5%) and achieved minimum dietary diversity (60%). The factors associated with complementary feeding were maternal education and occupation, and child characteristics such as birth order, and male gender [8]. Another study reported similar results i.e., only 57% of mothers initiated complementary feeding at the age of 6 months whereas 84% of children comply with minimum meal frequency and 35% of children comply with minimum meal diversity. Maternal education and having had their children's growth monitored were significantly associated with complementary feeding practices [9]. These studies studied the indicators of IYCF and associated factors individually. So, a compressive study is needed to assess the revised individual and composite indicators of IYCF and associated factors. The current study aims to identify the prevalence of Minimum Meal Frequency (MMF), Minimum Dietary Diversity (MDD), Minimum Acceptable Diet (MAD), Timely Introduction of Complementary Feeding, Appropriate Complementary Feeding Practice (ACFP), and the factors associated with Appropriate Complementary Feeding Practice.

## Methodology

### Study design and setting

From August 2018 to December 2018 a cross-sectional study was conducted to assess the different indicators of complementary feeding and factors associated with appropriate

complementary feeding practice in Bhimphedi Rural Municipality of Nepal. Bhimphedi rural municipality is one of nine municipalities of Makawanpur district of Bagmati province.

## Study population and sampling

The study population was the mothers of children aged 6–23 months who were permanent residents of Bhimphedi Rural Municipality or stayed at Bhimphedi Rural Municipality for at least one year. There was a total of 714 children aged 6–23 months at the start of the study. The eligible numbers of participants were enlisted from the HMIS 4.2 (FCHV Register) and HMIS 2.2 (Immunization Register) of each health facility in the study area. The mother's name of each child meeting the criteria was listed in the MS Excel 2016. The sample size was calculated by using the finite population correction formula with a 5% margin of error and 95% confidence level and taking the prevalence of minimum acceptable diet as 44% [7]. After adding a 5% non-response rate, the final sample size of 259 was calculated. The required sample size (n = 259) was drawn randomly by generating random numbers. The 259 research participants with the lowest random number were visited with the help of Female Community Health Volunteer (FCHV) and health workers for data collection.

## Measures

**Appropriate complementary feeding practice 6–23 months.** Appropriate complementary feeding practice is the dependent variable and it is measured by using a composite indicator comprising three of the WHO core IYCF indicators. These are the timely introduction of solid, complementary feeding, minimum dietary diversity, and minimum meal frequency [10]. If a child fulfilled all three criteria, it was classified as having received appropriate complementary feeding [11].

**Minimal diet diversity 6–23 months.** Minimal Diet Diversity for 6–23 months of age is the consumption of foods and beverages from at least five out of eight defined food groups during the previous day. The eight food groups are breast milk; grains, roots, tubers, and plantains; pulses (beans, peas, lentils), nuts and seeds; dairy products (milk, infant formula, yogurt, cheese); flesh foods (meat, fish, poultry, organ meats); eggs; vitamin-A rich fruits and vegetables; and other fruits and vegetables [10].

**Minimum meal frequency 6–23 months.** Minimum meal frequency is the proportion of children between 6–23 months of age who received solid, semi-solid, or soft foods for at least the minimum number of times recommended by WHO. Breastfed children aged 6–8 months and 9–23 months of age should consume solid or semi-solid food minimum twice a day and thrice a day, respectively. Non-breastfed children between 6 and 23 months of age should consume solid or semi-solid food at least four times a day, and also, they should also intake dairy or formula milk [10].

**Minimum acceptable diet 6–23 months.** For breastfeeding children, MAD is defined as children receiving at least the minimum dietary diversity and minimum meal frequency for their age during the previous day; And for non-breastfed children, MAD is defined as children receiving at least the minimum dietary diversity and minimum meal frequency for their age during the previous day as well as at least two milk feeds [10].

**Introduction of solid, semi-solid, or soft foods 6–8 months.** The proportion of infants 6–8 months of age who consumed solid, semi-solid, or soft foods during the previous day [10].

**Household food security.** Food security is defined as a state in which "all people at all times have both physical and economic access to sufficient food to meet their dietary needs for a productive and healthy life" [12]. Household food security was assessed through the Household Food Insecurity Access Scale (HFIAS). There are altogether nine questions. Each question

is asked with a recall period of four weeks. The respondent is first asked whether the condition in the question happened at all in the past four weeks. If the respondent answers "yes, a frequency-of-occurrence question is asked to determine whether the condition happened rarely (once or twice), sometimes (three to ten times), or often (more than ten times) in the past four weeks [13].

## Data collection

A structured questionnaire was used to collect the data from the research participants. The researcher himself was involved in the data collection and the process was closely supervised by an academic supervisor. The data were collected in a tablet phone-based questionnaire using the Open Data Kit (ODK) mobile application from 24 October 2018 to 5 November 2018.

## Data management and analysis

The collected data were assessed for completeness and consistency of information and uploaded to the Open Data Kit server on a daily basis. The compiled data were first exported to IBM SPSS version 21 for cleaning and analysis. The outcome variable was coded as '1' for appropriate complementary feeding practice and '0' for inappropriate complementary feeding practice. Descriptive statistical analysis was performed to compute the frequency, and percentage for nominal data and mean for continuous independent variables. Binary logistic regression was performed to determine the relationship of each independent variable with appropriate complementary feeding practice. Multivariable logistic regression was done to examine the relationship between independent variables and appropriate complementary feeding practice to address the confounding effect. Only those variables that were significant at a 5% level of significance in bivariate analysis were included in the multivariable logistic regression analysis. Hosmer and Lemeshow test was used to test the goodness-of-fit for regression models. The test statistic was 0.80 ($p > 0.05$) which showed that the model adequately fit the data.

## Ethical consideration

Ethical approval was taken from the Institutional Review Board (IRB) of the Institute of Medicine. Formal permission was also taken from the office of the respective rural municipality. The purpose of the study was clearly explained to the respondents and written consent was obtained from the study participants before collecting data. Those participants who were not practicing appropriate complementary feeding practice were provided with the correct information and advice was given to consult with Female Community Health Volunteers (FCHVs) and health workers at any point of service outlets during data collection.

## Results

Fig 1 shows the prevalence of indicators of complementary feeding of children aged 6–23 months. The overall prevalence of appropriate complementary feeding practice of mothers of children aged 6–23 months was 25%. Four out of five mothers had initiated complementary feeding to their children at the age of six to eight months. Seventy-one percent of the mothers met the WHO-recommended MMF, according to the age and breastfeeding status of the children. Similarly, 37% of mothers offer their children five or more types of food out of eight food groups. The prevalence of minimum acceptable diet for children aged 6–23 months was 27%.

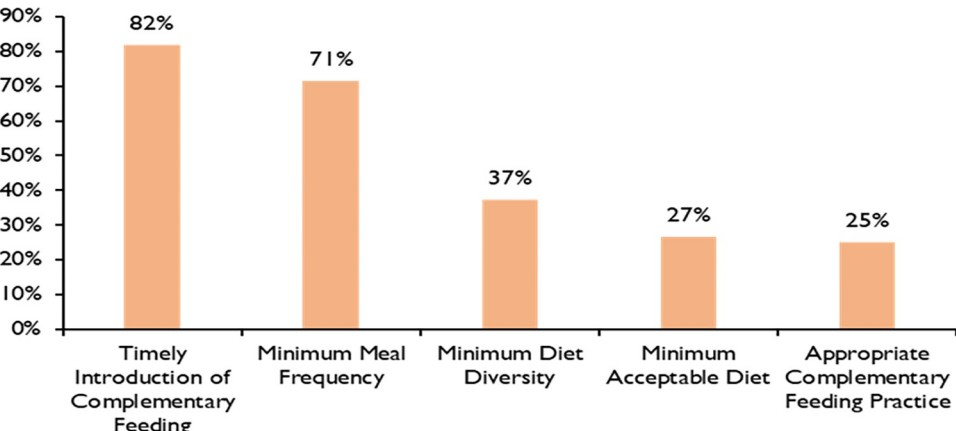

**Fig 1. Indicators of complementary feeding of children aged 6–23 months (n = 259).**

Table 1 depicts that the mean age of the mother and child was 27.1 years (SD 7.0 years) and 13.7 months (SD 4.7 months) respectively. Among the children majority (56.8%) were female. More than two-thirds (69.9%) of mothers had formal education whereas the remaining mothers were illiterate or literate only. Similarly, about three-fourths (73.7%) of fathers had attended formal education. Around 60% of the participants had five or fewer family members in their house. Out of the total respondents, 27.4% had paid jobs. Four-fifth (80.3%) of the participants had a family income below 32000 Nepalese rupees per month. Around three-fourths (73.0%) of mothers and their husbands decide on the type of food they eat. Sixty-three percent of the mothers had received counseling on IYCF. More than half (52.1%) of mothers had given birth to their child in the health facility. Only one-fourth of respondents had a kitchen garden in their homes. Sixty-eight percent of the families belonged to food-secured families.

Table 2 shows the bivariate and multivariate regression analysis of dependent and independent variables. In bivariate analysis, the education of the mother, education of the father, number of family members, monthly income of the family, counseling on IYCF, place of delivery, presence of kitchen garden, and household food security status were found to be associated with appropriate complementary feeding practice.

In multivariate analysis, mother and father with formal education (AOR 6.1, CI: 1.7–22.4 and AOR 5.6 CI: 1.5–21.2 respectively), counseling on IYCF (AOR 4.2, CI: 1.5–12.3), having kitchen garden (AOR 2.4, CI: 1.1–5.2) and food secured family (AOR 3.0, CI: 1.0–8.9) had positive association at 95% CI with the appropriate complementary feeding practice.

## Discussion

Appropriate complementary feeding during 6–23 months of life plays a crucial role in children's growth, development, and survival. This study finds that the appropriate complementary feeding practice was 25% which is lower than the study done in Oromia-Ethiopia 30% [14], North West Ethiopia 37.2% [15], and higher than the study done in Northern Ghana 15.7% [16], Southwestern Nigeria 4.2% [17], and South Ethiopia 8.5% [18]. This variation in the prevalence of appropriate complementary feeding practice might be attributed to the cultural, and socio-economic status and the time of information collection among countries.

In this study, 82% of mothers of children aged 6–23 had initiated complementary feeding at the age of 6–8 months. This finding is consistent with the findings from Nepal Demographic Health Survey 2016 [7], and higher than similar studies done in Ethiopia [11, 19] and

**Table 1. Socio-demographic, economic, and health service-related characteristics of the research participants.**

| Variable | Number (n = 259) | Percentage (%) |
|---|---|---|
| Age of mother | | |
| < 25 years | 104 | 40.2 |
| ≥ 25 years | 155 | 59.8 |
| Mean (SD) | | 27.1 (7.0) |
| Age of child | | |
| <12 months | 102 | 39.4 |
| ≥12 months | 157 | 60.6 |
| Mean (SD) | | 13.7 (4.7) |
| Sex of baby | | |
| Male | 112 | 43.2 |
| Female | 147 | 56.8 |
| Education of mother | | |
| Formal education | 181 | 69.9 |
| No formal education | 78 | 30.1 |
| Education of father | | |
| Formal education | 191 | 73.7 |
| No formal education | 68 | 26.3 |
| Number of family members | | |
| ≤ 5 | 154 | 59.5 |
| > 5 | 105 | 40.5 |
| Mean (SD) | | 5 (2) |
| Job of mother | | |
| Paid | 71 | 27.4 |
| Unpaid | 188 | 72.6 |
| Monthly income | | |
| < 32000 | 208 | 80.3 |
| ≥ 32000 | 51 | 19.7 |
| Median (lowest-highest) | | 20,000 (6,000–100,000) |
| Decision for food | | |
| Wife and husband | 189 | 73.0 |
| Other family member | 70 | 27.0 |
| Counselling on IYCF | | |
| Yes | 163 | 62.9 |
| No | 96 | 37.1 |
| Place of delivery | | |
| Home | 124 | 47.9 |
| HFs | 135 | 52.1 |
| Kitchen garden | | |
| Yes | 66 | 25.5 |
| No | 193 | 74.5 |
| Household food security status | | |
| Secure | 176 | 68.0 |
| Insecure | 83 | 32.0 |

Bangladesh [20]. Out of the research participants, 71% of the children received an appropriate frequency of complementary food as per the WHO recommendation. The finding of this study is consistent with the findings from the Nepal Demographic Health Survey 2016 [7], Southern Ethiopia [11]. However, the finding is higher than the study done in Ethiopia [21],

**Table 2. Factors associated with the appropriate complementary feeding practice.**

| Variables | Appropriate complementary feeding practice | | COR (95% CI) | AOR (95%CI) |
|---|---|---|---|---|
| | No (n = 194) | Yes (n = 65) | | |
| Education of mother | | | | |
| Formal education | 119 (65.7%) | 62 (34.3%) | 13.0 (3.9–43.0)* | 6.1 (1.7–22.4)* |
| No formal education | 75 (96.2%) | 3 (3.8%) | Ref | Ref |
| Education of father | | | | |
| Formal education | 129 (67.5%) | 62 (32.5%) | 10.4 (3.1–34.4)* | 5.6 (1.5–21.2)* |
| No formal education | 65 (95.6%) | 3 (4.4%) | Ref | Ref |
| Number of family members | | | | |
| ≤ 5 | 108 (70.1%) | 46 (29.9%) | 1.9 (1.1–3.5)* | 1.0 (0.5–2.2) |
| > 5 | 86 (81.9%) | 19 (18.1%) | Ref | Ref |
| Monthly family income | | | | |
| ≥ 32000 | 26 (51.0%) | 25 (49.0%) | 4.0 (2.1–7.7)* | 2.1 (0.9–4.9) |
| < 32000 | 168 (80.8%) | 40 (19.2%) | Ref | Ref |
| Counselling on IYCF | | | | |
| Yes | 103 (63.2%) | 60 (36.8%) | 10.6 (4.1–27.5)* | 4.2 (1.5–12.3)* |
| No | 91 (94.8%) | 5 (5.2%) | Ref | Ref |
| Place of delivery | | | | |
| Health facility | 89 (65.9%) | 46 (34.1%) | 2.9 (1.6–5.2)* | 0.9 (0.4–2.1) |
| Home | 105 (84.7%) | 19 (15.3%) | Ref | Ref |
| Kitchen garden | | | | |
| Yes | 39 (59.1%) | 27 (40.9%) | 2.8 (1.5–5.2)* | 2.4 (1.1–5.2)* |
| No | 155 (80.3%) | 38 (19.7%) | Ref | Ref |
| Household food security status | | | | |
| Secure | 116 (65.9%) | 60 (34.1%) | 8.1 (3.1–21.0)* | 3.0 (1.0–8.9)* |
| Insecure | 78 (94.0) | 5 (6.0%) | Ref | Ref |

*P<0.05

Northern Tanzania [22] and Bangladesh [20]. It was found that thirty-seven percent of mother offer at least five groups of food out of eight food groups to their children. This finding is higher than the study done in Ethiopia [21], Pakistan [23], and Bangladesh [20] and lower than Srilanka [24] and Oromia-Ethopia [14]. Similarly, 27% of children aged 6–23 months received a minimum acceptable diet. The finding is lower in comparison to the findings of NDHS-2016 [7], Oromia- Ethiopia [14], Bangladesh and Sirlanka [24]. It might be because most of the population relies on certain local staple foods like rice, wheat, potatoes, etc. Although children are fed at regular interval, the variety of food options is low. Furthermore, there is a pervasive cultural view in Nepal that cereal meals contain high energy and are sufficient for child growth, dismissing the value of a diverse diet. [9].

In multivariable logistic regression, maternal education shows a significant association with appropriate complementary feeding practice. Mothers who had attended formal education had higher odds of practicing appropriate complementary feeding practice than those who were illiterate or literate only (AOR 6.1, CI: 1.7–22.4). This finding is consistent with similar studies conducted in Ethiopia and Uganda [11, 25]. This could be due to educated mothers have greater confidence, a higher position within the household, and more ability to allocate household resources on their own compared to mothers with no schooling [26]. Educated mothers also can understand and internalize the behavior change communication message regarding complementary feeding and apply it in their daily life.

The involvement of the father is equally important while taking care of the child. Taking into consideration of patriarchal nature of Nepalese society men are conventionally considered responsible for major decisions within the households on many occasions as compared to their female counterparts [27]. In this situation, educated husbands can better understand and support their wives by approving what mothers would like to do to keep their children healthy. Educated husbands might enhance the wife's awareness of the timely initiation of complementary feeding to their infant [19]. Like the mother, an educated father also can understand and internalize the behavior change communication message regarding complementary feeding and support the mother to comply with appropriate complementary feeding practice. This study showed a positive association between the father's education and appropriate complementary feeding practice (AOR 5.6, CI: 1.5–21.2). This finding is supported by similar studies done in China [28] and Southwest Ethiopia [19].

Comprehensive information regarding complementary feeding motivates mothers to practice appropriate complementary feeding by increasing their knowledge regarding IYCF and helping to overcome the difficulties. In Nepal, counseling on IYCF was done during childbirth, PNC visits, growth monitoring, and immunization. The repeated information will encourage mothers to follow the appropriate complementary feeding practice as per the age of the baby. Even FCHV provides mothers with information regarding IYCF as per the age of the baby. This study identified that the mothers who received counseling on IYCF had higher odds of practicing appropriate complementary feeding (AOR: 4.2, CI: 1.5–12.3) in compression to those who did not receive any counseling. This finding is supported by studies done in Ethiopia [14, 15, 29], Nepal [9], and Nigeria [30].

The best place to find an array of fresh fruits and vegetables is the kitchen garden. The kitchen garden plays a crucial role in ensuring diverse food groups at the table as recommended by WHO. This study revealed that those families who had kitchen garden were two times more likely to follow appropriate complementary feeding practice in comparison to those who did not have kitchen garden (AOR 2.4, CI: 1.1–5.2).

Mothers who are from food-secure households were more likely to practice appropriate complementary feeding practice in compression to those who were from food-insecure house households (AOR 3.0, CI: 1.0–8.9). This finding is consistent with the study done in Ethiopia [14, 31, 32] and Bangladesh [20, 33]. This can be explained by the fact that food secured family has access to nutritious food throughout the year.

Some limitations need to be considered when interpreting the results of this study. One of the important limitations is recall bias because the information on IYCF indicators was collected on a single 24-hour recall basis. Seasonal variation and cultural factors were not assessed in this study which might have affected the type of food consumed by the infant and young child. Due to the cross-sectional nature of this study, conclusions on the cause-effect relationship cannot be drawn.

## Conclusion

This study revealed that the prevalence of MDD and MAD were low. Similarly, a significant proportion of mothers had inappropriate complementary feeding practice for their children aged 6–23 months. Factors such as spouse education, counseling on IYCF, kitchen garden, and household food security had positive association with appropriate complementary feeding practice. To improve the appropriate complementary feeding practice, this study highlights the need for behavior change communication interventions for illiterate or literate-only parents. Similarly, efforts should be made to promote food diversity and food security to increase access to nutritious food for families. Encouraging the establishment of kitchen garden can be an effective strategy to improve MDD and complementary feeding practices.

## Supporting information

**S1 File.**
(SAV)

## Acknowledgments

Authors would like to express our gratitude to the faculty members of the Central Department of Public Health and Department of Community Medicine, Institute of Medicine, Tribhuvan University, Nepal, for their invaluable guidance throughout the course of our research project. We thank Mr. Dhruba Prasad Lamichhane, Health Coordinator of Bhimphedi rural municipality and Health Assistant, Mr. Dipesh Shrestha from Ipa Health Post for their support in establishing coordination at study sites, as well as to FCHV, and local health workers for their support during data collection. The authors would also like to express sincere thanks to all research participants for their valuable time and information.

## Author Contributions

**Conceptualization:** Thag Bahadur Gurung, Rajan Paudel.

**Data curation:** Thag Bahadur Gurung, Anil K. C., Aashish Acharya, Pravin Kumar Khanal.

**Formal analysis:** Thag Bahadur Gurung, Anil K. C., Aashish Acharya, Pravin Kumar Khanal.

**Investigation:** Thag Bahadur Gurung.

**Methodology:** Thag Bahadur Gurung, Rajan Paudel.

**Supervision:** Rajan Paudel.

**Visualization:** Anil K. C.

**Writing – original draft:** Thag Bahadur Gurung, Anil K. C.

**Writing – review & editing:** Thag Bahadur Gurung, Rajan Paudel, Anil K. C., Aashish Acharya, Pravin Kumar Khanal.

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
