## [Decision Letter · Decision Letter 0]

17 Nov 2023

PONE-D-23-31826Appropriate complementary feeding practice and associated factors among mothers of children aged 6-23 months in Bhimphedi rural municipality of NepalPLOS ONE

Dear Dr. K.C.,

Thank you for submitting your manuscript to PLOS ONE. After careful consideration, we feel that it has merit but does not fully meet PLOS ONE’s publication criteria as it currently stands. Therefore, we invite you to submit a revised version of the manuscript that addresses the points raised during the review process. There are minor comments from the reviewers. Kindly address those to strengthen your manuscript on technical terms.

We look forward to receiving your revised manuscript.

Kind regards,

Naveed Sadiq, Ph.D.

Academic Editor

PLOS ONE

3. We are unable to open your Supporting Information file [Supporting Information.sav]. Please kindly revise as necessary and re-upload.

Reviewers' comments:

Reviewer's Responses to Questions

**Comments to the Author**

1. Is the manuscript technically sound, and do the data support the conclusions?

Reviewer #1: Yes

Reviewer #2: Yes

2. Has the statistical analysis been performed appropriately and rigorously? 

Reviewer #1: Yes

Reviewer #2: Yes

3. Have the authors made all data underlying the findings in their manuscript fully available?

Reviewer #1: Yes

Reviewer #2: Yes

4. Is the manuscript presented in an intelligible fashion and written in standard English?

Reviewer #1: No

Reviewer #2: Yes

5. Review Comments to the Author

Reviewer #1: 1. Line (L) 61-62: WHO recommended to continued breastfeeding up to 2 years of beyond not 12 months as stated. Any comment or justification on author's statement on this matter?

2. L61-65: The statement is not clear. Please paraphrase the statement to be clearer and what are on basis of this statement?

3. L88-89: I would suggest the author to consider to follow and standardised the indicator's name according to WHO Unicef IYCF indicator 2021. 'Introduction of solid, semi-solid or soft food 6-8 months (ISSSF)

4. L113-119: I would recommend the author to elaborate more on the selection of 3 indicators from 17 IYCF WHO IYCF practice's indicators. To justify why choose the three selected indicators and how classified it as Appropriate Complementary Feeding Practices. The term of 'core indicator' was used in WHO IYCF Practices Indicators 2008 and not in WHO IYCF Practices Indicators 2021 as the revised version divided into 3 main indicators (Breastfeeding, Complementary Feeding and Other practices indicator) rather than 2 main indicator in 2008 (Core vs Optional indicators)

As ISSSF become one of the three indicator to determine prevalence of ACFP, how the author analyse data as the ISSSF data is meant for the children aged 6-8 months (age group to assess the indicator). Can the author explain on how author assess this indicator? If the denominator was the children age 6-23 months, author need to reanalyse the data.

5. L141-143: Rephrase in the better way to explain the definition of the topic.

6. L239-240: The indicator was not follow the definition of WHO IYCF Practices Indicator 2021 (ISSSF). If the author do not want to follow the indicator's definition, the author need to explain and elaborate more on this topic.

Figure 1: Fig 1 showed the ISSSF indicator data come from the total respondent which is not follow the WHO guideline to asses the IYCF indicators. I assumed the author used the definition as states at line 141-143 and using the WHO IYCF Indicators for assessing infant and young child feeding practices. Standardise in term of definition is important especially to compare the result among the research on this topic.

All the best

Thank you

Reviewer #2: 1. As per the measure in methodology section, appropriate complementary feeding practice is defined as indicator three of the WHO core IYCF i.e. timely introduction of solid, complementary feeding minimum dietary diversity and minimum meal frequency. But in the result section, appropriate complementary feeding practice and all these indicators presented separately. So be clear about the measurement. If you need any correction, correct both in the abstract result section and main findings.

2. Better to specify the "to address associated factors" in conclusion section of abstract.

3. What is basis for classification of monthly income? Better use any standard basis

4. Found missing figure 1 in results section

5. it would be more clear for reader if you include fig. or table related to descriptive findings i.e. timing of introduction of solid, complementary feeding minimum dietary diversity and minimum meal frequency in results section.

6. PLOS authors have the option to publish the peer review history of their article (what does this mean?). If published, this will include your full peer review and any attached files.

Reviewer #1: **Yes: **Khairul Hasnan Amali

Reviewer #2: **Yes: **Gayatri Rai

---

## [Author Response · Author response to Decision Letter 0]

24 Nov 2023

Response: Rechecked

Response: Full ethics statement is included in method section under Ethical Consideration section.

3. We are unable to open your Supporting Information file [Supporting Information.sav]. Please kindly revise as necessary and re-upload.

Response: Supporting Information file is SPSS data file. I have re-uploaded that file. I have also shared that file on figshare https://doi.org/10.6084/m9.figshare.24633153.v1.

Response: Checked. We have not cited papers that have been retracted.

Review Comments to the Author

Reviewer #1: 1. Line (L) 61-62: WHO recommended to continued breastfeeding up to 2 years of beyond not 12 months as stated. Any comment or justification on author's statement on this matter?

Response: This is the estimated contribution of breastfeeding (Exclusive breastfeeding in the first 6 months of life and continued breastfeeding from 6 to 11 months) and complementary feeding on the prevention of under five deaths.

2. L61-65: The statement is not clear. Please paraphrase the statement to be clearer and what are on basis of this statement?

Response: Paraphrased

3. L88-89: I would suggest the author to consider to follow and standardised the indicator's name according to WHO Unicef IYCF indicator 2021. 'Introduction of solid, semi-solid or soft food 6-8 months (ISSSF)

Response: “Complementary feeding” indicator is replaced with “Introduction of solid, semi-solid or soft food 6-8 months (ISSSF)”

4. L113-119: I would recommend the author to elaborate more on the selection of 3 indicators from 17 IYCF WHO IYCF practice's indicators. To justify why choose the three selected indicators and how classified it as Appropriate Complementary Feeding Practices. The term of 'core indicator' was used in WHO IYCF Practices Indicators 2008 and not in WHO IYCF Practices Indicators 2021 as the revised version divided into 3 main indicators (Breastfeeding, Complementary Feeding and Other practices indicator) rather than 2 main indicator in 2008 (Core vs Optional indicators)

As ISSSF become one of the three indicator to determine prevalence of ACFP, how the author analyse data as the ISSSF data is meant for the children aged 6-8 months (age group to assess the indicator). Can the author explain on how author assess this indicator? If the denominator was the children age 6-23 months, author need to reanalyse the data.

Response: Necessary changes were made.

The research participants of this study were the mothers of children of age 6-23 months. The information regarding ISSSF was collected retrospectively from all the respondent’s whether they initiated introduction of solid, semi-solid or soft food at the age of 6-8 month. The denominator in this indicator contains the information of ISSSF when the age of the children was 6-8 months.

5. L141-143: Rephrase in the better way to explain the definition of the topic.

Response: Previous operation definition is changed to “The proportion of children age 6–23 months who had been introduced to solid, semi-solid, or soft foods at the age between 6 to 8 months.” 

6. L239-240: The indicator was not follow the definition of WHO IYCF Practices Indicator 2021 (ISSSF). If the author do not want to follow the indicator's definition, the author need to explain and elaborate more on this topic.

Response: Operation definition of ISSSF is revised to “The proportion of children age 6–23 months who had been introduced to solid, semi-solid, or soft foods at the age between 6 to 8 months”

Figure 1: Fig 1 showed the ISSSF indicator data come from the total respondent which is not follow the WHO guideline to asses the IYCF indicators. I assumed the author used the definition as states at line 141-143 and using the WHO IYCF Indicators for assessing infant and young child feeding practices. Standardise in term of definition is important especially to compare the result among the research on this topic.

Response: Operation definition was changed. Even though, almost all study cited the standard definition of WHO guideline, they have calculated ISSSF by taking total respondents, so we can compare the result with these studies.

Reviewer #2: 1. As per the measure in methodology section, appropriate complementary feeding practice is defined as indicator three of the WHO core IYCF i.e. timely introduction of solid, complementary feeding minimum dietary diversity and minimum meal frequency. But in the result section, appropriate complementary feeding practice and all these indicators presented separately. So be clear about the measurement. If you need any correction, correct both in the abstract result section and main findings.

Response: All three indicators along with Appropriate complementary feeding practice was presented in the figure because those indicator has equal importance in Infant and Young Child Nutrition and will also give readers comprehensive picture from which ACFP is derived.

2. Better to specify the "to address associated factors" in conclusion section of abstract.

Response: Associated factors are presented in result section.

3. What is basis for classification of monthly income? Better use any standard basis

Response: Monthly income was not classified in any standard basis because authentic data related to average monthly income of general population is lacking. 

4. Found missing figure 1 in results section

Response: Figure was uploaded separately

5. it would be more clear for reader if you include fig. or table related to descriptive findings i.e. timing of introduction of solid, complementary feeding minimum dietary diversity and minimum meal frequency in results section.

Response: As per the guideline of PLOS ONE, all figures should be uploaded separately. So, figure was uploaded separately.

---

## [Decision Letter · Decision Letter 1]

6 Feb 2024

PONE-D-23-31826R1Appropriate complementary feeding practice and associated factors among mothers of children aged 6-23 months in Bhimphedi rural municipality of NepalPLOS ONE

Dear Dr. K.C.,

Thank you for submitting your manuscript to PLOS ONE. After careful consideration, we feel that it has merit but does not fully meet PLOS ONE’s publication criteria as it currently stands. Therefore, we invite you to submit a revised version of the manuscript that addresses the points raised during the review process. The abstract is a bit unclear. Please make necessary changes as per the reviewer comments.

We look forward to receiving your revised manuscript.

Kind regards,

Naveed Sadiq, Ph.D.

Academic Editor

PLOS ONE

Journal Requirements:

Reviewers' comments:

Reviewer's Responses to Questions

**Comments to the Author**

1. If the authors have adequately addressed your comments raised in a previous round of review and you feel that this manuscript is now acceptable for publication, you may indicate that here to bypass the “Comments to the Author” section, enter your conflict of interest statement in the “Confidential to Editor” section, and submit your "Accept" recommendation.

Reviewer #1: All comments have been addressed

Reviewer #2: All comments have been addressed

2. Is the manuscript technically sound, and do the data support the conclusions?

Reviewer #1: Yes

Reviewer #2: Yes

3. Has the statistical analysis been performed appropriately and rigorously? 

Reviewer #1: Yes

Reviewer #2: Yes

4. Have the authors made all data underlying the findings in their manuscript fully available?

Reviewer #1: Yes

Reviewer #2: Yes

5. Is the manuscript presented in an intelligible fashion and written in standard English?

Reviewer #1: Yes

Reviewer #2: Yes

6. Review Comments to the Author

Reviewer #1: May i suggest to the author to remove Lines 229-230 as it is repetitive?

And Good luck to the author

Reviewer #2: Authors probably misunderstood the previous round comment 1. I think the study's outcome variable is appropriate complementary feeding practice (ACFP). They have been measured ACFP by using a composite indicator comprising three of the WHO core IYCF indicators i.e. timely introduction of solid, complementary feeding, minimum dietary diversity, and minimum meal frequency. But in abstract results are not so clear therefore, kindly please go through the main finding's result section and write in that way. I accept that the three indicators are equal important but the comment is just for writing style and clarity.

7. PLOS authors have the option to publish the peer review history of their article (what does this mean?). If published, this will include your full peer review and any attached files.

Reviewer #1: **Yes: **Khairul Hasnan Amali

Reviewer #2: No

---

## [Author Response · Author response to Decision Letter 1]

10 Feb 2024

Reviewer #1: May i suggest to the author to remove Lines 229-230 as it is repetitive?

And Good luck to the author

Response: Removed

Thank you

Reviewer #2: Authors probably misunderstood the previous round comment 1. I think the study's outcome variable is appropriate complementary feeding practice (ACFP). They have been measured ACFP by using a composite indicator comprising three of the WHO core IYCF indicators i.e. timely introduction of solid, complementary feeding, minimum dietary diversity, and minimum meal frequency. But in abstract results are not so clear therefore, kindly please go through the main finding's result section and write in that way. I accept that the three indicators are equal important but the comment is just for writing style and clarity.

Response: Necessary changes have been made in abstract

---

## [Editor Report · Decision Letter 2]

21 Feb 2024

Appropriate complementary feeding practice and associated factors among mothers of children aged 6-23 months in Bhimphedi rural municipality of Nepal

PONE-D-23-31826R2

Dear Dr. K.C.,

We’re pleased to inform you that your manuscript has been judged scientifically suitable for publication and will be formally accepted for publication once it meets all outstanding technical requirements.

Kind regards,

Naveed Sadiq, Ph.D.

Academic Editor

PLOS ONE
---

## [Editor Report · Acceptance letter]

25 Feb 2024

PONE-D-23-31826R2 

PLOS ONE

Dear Dr. K.C., 

I'm pleased to inform you that your manuscript has been deemed suitable for publication in PLOS ONE. Congratulations! Your manuscript is now being handed over to our production team.

Kind regards, 

on behalf of

Dr. Naveed Sadiq 

Academic Editor

PLOS ONE